# Availability of personal protective equipment and diagnostic and treatment facilities for healthcare workers involved in COVID-19 care: A cross-sectional study in Brazil, Colombia, and Ecuador

Jimmy Martin-Delgado[1,¤]*, Eduardo Viteri[2,3], Aurora Mula[1], Piedad Serpa[4], Gloria Pacheco[4‡], Diana Prada[4‡], Daniela Campos de Andrade Lourenção[5,6‡], Patricia Campos Pavan Baptista[5,6‡], Gustavo Ramirez[7‡], Jose Joaquin Mira[1,8,9]

1 Atenea Research Group, Foundation for the Promotion of Health and Biomedical Research, Sant Joan d'Alacant, Spain, 2 Santander Ophthalmologic Foundation FOSCAL, Floridablanca, Colombia, 3 CEMEDIP, Guayaquil, Ecuador, 4 Santander University, Bucaramanga, Colombia, 5 Sao Paulo University, Sao Paulo, Brazil, 6 CNPQ research group, Studies on the health of nursing and health workers, Sao Paulo, Brazil, 7 Catholic University of Santiago of Guayaquil, Guayaquil, Ecuador, 8 Alicante-Sant Joan Health Department, Alicante, Spain, 9 Miguel Hernandez University, Elche, Spain

☯ These authors contributed equally to this work.
¤ Current address: Sant Joan University Hospital, Sant Joan, Alacant, Spain
‡ These authors also contributed equally to this work.
* jimmy.martind@umh.es

## Abstract

Many affected counties have had experienced a shortage of personal protective equipment (PPE) during the coronavirus disease (COVID-19) pandemic. We aimed to investigate the needs of healthcare professionals and the technical difficulties faced by them during the initial outbreak. A cross-sectional web-based survey was conducted among the healthcare workforce in the most populous cities from three Latin American countries in April 2020. In total, 1,082 participants were included. Of these, 534 (49.4%), 263 (24.3%), and 114 (10.5%) were physicians, nurses, and other professionals, respectively. At least 70% of participants reported a lack of PPE. The most common shortages were shortages in gown coverall suits (643, 59.4%), N95 masks (600, 55.5%), and face shields (569, 52.6%). Professionals who performed procedures that generated aerosols reported shortages more frequently (p<0.05). Professionals working in the emergency department and primary care units reported more shortages than those working in intensive care units and hospital-based wards (p<0.001). Up to 556 (51.4%) participants reported the lack of sufficient knowledge about using PPE. Professionals working in public institutions felt less prepared, received less training, and had no protocols compared with their peers in working private institutions (p<0.001). Although the study sample corresponded to different hospital centers in different cities from the participating countries, sampling was non-random. Healthcare professionals in Latin America may face more difficulties than those from other countries, with 7 out of 10 professionals reporting that they did not have the necessary resources to care for patients

**Data Availability Statement:** Raw data files are available from the OSF database (https://doi.org/10.17605/OSF.IO/4EG8R).

**Funding:** This study used research funding of the Atenea Research Group of the Foundation for the Promotion of Health and Biomedical Research – FISABIO. It did not receive any other funding from public or private institutions. On the other hand, CEMEDIP did not provided support to this study. CEMEDIP provided in the form of partial-time salaries for author EV. CEMEDIP did not have any additional role in the study design, data collection and analysis, decision to publish, or preparation of the manuscript. EV participated in conception and design of the survey. The specific roles of these authors are articulated in the 'authors contributions' section.

**Competing interests:** Authors have declared no competing interest exist. This does not alter our adherence to PLOS ONE policies on sharing data and materials. CEMEDIP is a private institution based in Ecuador which provides academic training to healthcare related professionals and students. It has not been part of conception, design, data analysis or drafting the manuscript of this research. EV is part of CEMEDIP as partial-time professor for which he receives a salary. EV principal affiliation is with Santander Ophthalmologic Foundation FOSCAL where he currently in undergoing a fellow. We have updated the author affiliation. We attest that we have no commercial associations (e.g., equity ownership or interest, consultancy, patent and licensing agreements, or institutional and corporate associations) that might be a conflict of interest in relation to the submitted manuscript. All sources of funding in support of the work presented in the article are indicated.

with COVID-19. Technical and logistical difficulties should be addressed in the event of a future outbreak, as they have a negative impact on healthcare workers.

**Clinical trial registration:** NCT04486404

## Introduction

On December 31, 2019, several cases of pneumonia of unknown etiology in Wuhan were reported by the People's Republic of China to the World Health Organization (WHO). Later, the causative agent was found to be a novel coronavirus, which was subsequently called severe acute respiratory coronavirus 2 (SARS-CoV-2). Infection with SARS-CoV-2 can result in coronavirus disease (COVID-19), which presents with respiratory and other symptoms [1]. On March 11, 2020, the WHO declared the COVID-19 outbreak to be a global pandemic [2].

On February 26, 2020, the first case of COVID-19 was registered in South America [3]. Since then, the number of confirmed cases has increased to 8,703,722, with 272,278 deaths (as of October 14, 2020). Brazil, Ecuador, and Colombia are among the most affected countries worldwide [4].

Healthcare networks in the most affected cities have exceeded their operational capacity [5]. This has highlighted a number of challenges posed by the excessive demand for care, hospitalization, intensive care, and management of patients suspected or confirmed to have COVID-19. Furthermore, special biosecurity and protection measures are required to protect the healthcare workforce.

Several studies have demonstrated that healthcare workers worldwide have been facing an overwhelming workload, lack of personal protection equipment (PPE), lack of ventilators and drugs, and a feeling of inadequate support due to the COVID-19 pandemic [6, 7]. This situation has generated deep concern among health and administrative workers, with some institutional measures being implemented to counteract these concerns. Additionally, the availability of PPE and of protocols standardizing its proper use vary. Due to an increased perception of self-risk (according to recent data, the perceived risk is up to two times higher than that in the general population), the demand for PPE is high [8].

In a worldwide survey, only 3% of the 2,711 included healthcare workers were from South America, and up to 52% of all participants reported the unavailability of at least one piece of standard PPE [9]. For instance, in the United States, the Food and Drug Administration and the Centers for Disease Control and Prevention have adopted several measures to optimize PPE use due to its shortage [10, 11]. Even though professionals consider that they have been well prepared for the pandemic in Germany, substantial differences in PPE availability have still been reported, depending on the setting (ambulatory or maximum-care hospitals) [12]. In Spain, 54% of primary care healthcare workers have not been adequately trained regarding how to use PPE [13]. A recent study found that the use of standardized PPE, including protective suits, masks, gloves, goggles, face shields, and gowns, could reduce the risk of contagion [14].

PPE shortages and the lack of preparedness have been reported as common issues in most affected countries [15]. However, the healthcare systems in South America are weaker than those in regions with stronger economies and more healthcare funding. No studies have considered this issue, and there is no data on PPE availability, diagnostic testing of professionals, and training in South American countries during the COVID-19 pandemic.

This study aimed to examine PPE shortages and the level of preparedness in South America from the perspective of healthcare professionals in South American countries [16]. In addition, this study aimed to examine the training and other needs of healthcare workers and the technical difficulties faced by them during the initial outbreak.

## Materials and methods

A cross-sectional study was conducted during the first phase of the pandemic among the healthcare workforce from Brazil, Colombia, and Ecuador. Participation was voluntary and anonymous. This study was carried out in accordance with the Declaration of Helsinki and the normative and ethical regulations of the participating countries. The participants provided online informed consent prior to the survey. The ACCADEMY group guide for self-administered surveys of clinicians was followed [17].

### Survey instrument

A survey was developed by a focus group using virtual communication channels. This technique has been used previously in other studies with satisfactory results and was primarily used due to social distancing requirements [18, 19]. Medical doctors from different hospitals in the city of Guayaquil, Ecuador, were included. A list of difficulties faced in patient care during the COVID-19 outbreak was compiled at the meeting. This was later checked by the research teams in Colombia and Brazil and adapted cross-culturally.

These emergent themes were used to create a digital survey with different questions, including multiple-choice questions, questions involving the listing of priorities, and open questions, to obtain qualitative information. There were no personal questions or questions regarding site-specific work conditions to preserve privacy and ensure the protection of personal data. In addition, the option to prevent the input of duplicate answers was enabled.

The survey content was submitted to a discussion group made up of six physicians; this group was different from the focus group in Ecuador. Modifications and changes were proposed by this group to improve the questions, reduce errors, and improve legibility. Furthermore, the content and readability were checked by two researchers in Colombia. The questions were translated and cross-culturally adapted following Beaton's recommendations [20]. This same process was also undertaken by two other researchers who were fluent in Spanish and Portuguese; then, the content and readability were checked by a small group of Brazilian healthcare workers.

The survey had 14 questions, and the completion time was 6 minutes on average. Themes and obstacles were identified and grouped according to the following emergent themes: the number of staff; PPE (following WHO recommendations [21] for different levels of exposure or procedures involved), resources for appropriate patient treatment, availability of equipment, COVID-19 protocols, handling of personnel with suspected infection, and training (S1 File).

### Participants

According to the official statistics, as of 2019 (latest available data), there are 82,009, 168,810, and 691,350 healthcare professionals (including physicians and nurses) in Ecuador, Colombia, and Brazil, respectively. Therefore, there was an estimated pool of 942,169 healthcare professionals [22, 23]. The sample size of 829 respondents was defined using the formula for infinite universes, with a 99% confidence level, 5% accuracy (p = q = 50), and 20% of lost data. At least 340 participants were required from each country for the sample to be representative. The target population of the survey was healthcare workers of any discipline or training background who were caring for patients with COVID-19. These healthcare workers included physicians

(medical doctors who have completed a specialization, are undergoing training for a specialization, or are working as general doctors), nurses, auxiliary nurses, and other professionals (psychologists, physiotherapists, and respiratory therapists). Participants were required to specify their area of work and to specify whether the institution was a part of the public health system or a private hospital. Healthcare workers from different cities in each country were invited to participate so as to include a more representative sample. Participants were divided into two groups: performance of aerosol-generating (those working in intensive care units, general wards, and emergency departments) and no performance of aerosol-generating procedures (those providing primary care and radiology services). This categorization was performed as aerosol-generating procedures are associated with an increased risk of infection and normally require additional PPE.

## Survey administration

We used Surveymonkey®, a web-based survey platform (SurveyMonkey Inc., San Mateo, USA), which prohibited the duplication of answers using internet protocol address information. A non-random, purposive sample of participants was invited from April 4 to May 7, 2020, using an e-mail database of over 3,000 people, social media, and instant messaging applications. Data collection continued until an adequate number of healthcare workers from each of the participating countries had been surveyed for the sample to be considered representative.

## Data management and analysis

Descriptive analysis was conducted using IBM SPSS Statistics (IBM Corp. Released 2017. IBM SPSS Statistics for Windows, Version 25.0. Armonk, NY: IBM Corp). The results for each item were reported according to each participating country or according to different healthcare professions. An inferential analysis was conducted using the chi-square test to compare specific variables (public and private hospitals, exposure to high-risk procedures, and professional categories). A p-value of $<0.05$ was considered statistically significant (confidence intervals at 95%). Comments were extracted from the open text questions in the survey and analyzed by theme, most frequent narratives, and participants' perceptions of other obstacles.

## Results

In the study period, 1,153 responses were collected, but only 1,082 were considered valid and included in the analysis. Of the 1,082 participants, 352 (32.5%), 389 (36%), and 341 (31.5%) were from Brazil, Colombia, and Ecuador, respectively. Overall, 538 (49.7%), 273 (25.4%), and 145 (13.4%) participants worked in public hospitals, private hospitals, and primary care units, respectively. Overall, 324 (30%), 295 (27.3%), and 278 (25.7%) worked in a hospital emergency department, hospital ward, and hospital primary care unit, respectively. A total of 755 (70.4%) participants worked in areas where aerosol-generating procedures were performed. In Ecuador and Colombia, the sample consisted mostly of physicians; in contrast, the sample in Brazil consisted mostly of nurses (Table 1).

## Resources for appropriate diagnosis and treatment

In total, 756 (70%) participants reported a lack of resources for diagnosing and treating patients with COVID-19. Emergency department and primary care staff reported greater shortages in medicines and equipment for diagnosing and treating patients with COVID-19 than those working in hospital-based wards and intensive care units (ICUs; S1 Table).

**Table 1. Participants description.**

| | Brazil | | Colombia | | Ecuador | |
|---|---|---|---|---|---|---|
| | n = 352 | | n = 389 | | n = 341 | |
| **Type of institution** | n | % | n | % | n | % |
| Public institutions | 281 | 79.8 | 134 | 34.4 | 268 | 78.6 |
| Private institutions | 71 | 20.2 | 255 | 65.6 | 73 | 21.4 |
| **Professional group** | | | | | | |
| Physician | 2 | 0.6 | 238 | 61.2 | 294 | 86.2 |
| Nurse | 191 | 54.3 | 54 | 13.9 | 18 | 5.3 |
| Nursing Assistant | 145 | 41.2 | 23 | 5.9 | 3 | 0.9 |
| Others | 14 | 4.0 | 74 | 19.0 | 26 | 7.6 |
| **Work area** | | | | | | |
| Emergencies | 75 | 21.3 | 104 | 26.7 | 145 | 42.5 |
| Hospital ward | 91 | 25.9 | 126 | 32.4 | 78 | 22.9 |
| Intermediate or Intensive Care Unit | 40 | 11.4 | 63 | 16.2 | 33 | 9.7 |
| Radiology services | 9 | 2.6 | 20 | 5.1 | 10 | 2.9 |
| Primary care | 137 | 38.9 | 76 | 19.5 | 65 | 19.1 |
| **Works in an aerosol-generating service** | | | | | | |
| Presence | 206 | 58.5 | 293 | 75.3 | 256 | 75.1 |
| Absence | 146 | 41.5 | 96 | 24.7 | 75 | 22.0 |

Professionals from public institutions who worked in areas where aerosol-generating procedures were performed reported higher resource shortages than those working in private institutions (p<0.001; Table 2).

## PPE

Only 145 (13.4%) participants considered that they had adequate PPE for treating patients with COVID-19. In particular, 643 (59.4%), 600 (55.5%), and 569 (52.6%) participants reported shortages of gown coveralls, N95 masks, and face shields, respectively (S2 Table).

Among participants who worked in areas where aerosol-generating procedures were performed, 448 (59.3%), 384 (50.9%), and 372 (49.3%) reported the lack of special closed suits, N95 type masks, and face protectors, respectively. In total, 92 (29%) professionals who did not

**Table 2. Lack of resources in the opinion of professionals working in areas of activity that perform procedures that generate aerosols, according to the type of institution.**

| | Public institutions n = 460/755 (60,9%) | | Private institutions n = 295/755 (39,1%) | | p-Value |
|---|---|---|---|---|---|
| | n | % | n | % | |
| **In general, during your previous workdays, during the care of a patient with possible respiratory affection, did you modify your therapeutic or diagnostic behavior for any of the following reasons? You can choose more than one.** | | | | | |
| **Unavailability of necessary medication** | 141 | 30.7 | 49 | 16.6 | <0.001 |
| **Lack of access to non-invasive ventilatory support (oxygen, cannulas, humidifiers, masks)** | 117 | 25.4 | 45 | 15.3 | 0.001 |
| **Lack of access to intensive care or invasive mechanical ventilation (ventilator)** | 178 | 38.7 | 61 | 20.7 | <0.001 |
| **Lack of access to necessary diagnostic imaging tests** | 129 | 28.0 | 47 | 15.9 | <0.001 |
| **Lack of access to necessary laboratory tests** | 177 | 38.5 | 75 | 25.4 | <0.001 |
| **I have had the necessary to diagnose/treat patients** | 103 | 39.8 | 137 | 57.1 | <0.001 |

Only participants working in areas where aerosols were generated were included in this analysis (n = 755).

perform aerosol-generating procedures said they did not have surgical masks. Only 109 (21.8%) participants who performed aerosol-generating procedures and 36 (14.9%) participants who did not perform such procedures said they had the necessary equipment to adequately care for patients with COVID-19 (Table 3).

Up to 141 (13%) participants reported that they had to supply their own PPE (obtained through personal means). Furthermore, 528 (48.8%) participants stated that they reused PPE after it had been sterilized by themselves or at their workplace. The rest of the participants reported that they disposed off the PPE after use.

## Obstacles faced by healthcare workers

The lack of diagnostic tests and PPE were prioritized (on a scale of 0–10) by the participants as the main obstacles faced while caring for patients with COVID-19. Fig 1 details the other obstacles reported by the participants.

## Training on how to use PPE and awareness of relevant protocols

Up to 556 (51.4%) participants reported that they had not been trained on the correct use of PPE. Of these participants, 360 (64.7%) worked in public institutions and 196 (35.3%) worked in private institutions (p<0.001).

In total, 996 (92%) participants acknowledged that a protocol was used at their workplace to care for patients with COVID-19. The national guidelines were the most frequently used (570, 52.7%), followed by protocols developed by the institution (226, 20.9%). However, the technical, logistical, and inventory-related limitations inherent to the workplace comprised the main difficulties encountered in implementing these protocols (476, 44%). Moreover, 291 (26.9%) participants were unaware of whether such protocols existed (Table 4). Professionals working in public institutions considered themselves less prepared than those working in private institutions (p<0.001).

## Handling of personnel with suspected or confirmed infection

In the scenario that a frontline professional presented with symptoms associated with COVID-19, 632 (58.4%) participants reported that their workplace arranged for reverse transcriptase-

**Table 3. Lack of personal protective equipment according to whether they performed procedures that generated aerosols.**

| | Performed n = 755/1072 (70,4%) | | Did not perform n = 317/1072 (29,6%) | | p-value |
|---|---|---|---|---|---|
| | n | % | n | % | |
| **In general, what protective equipment was needed to care for patients suspected of having respiratory problems during your shift or care days, and have you been unable to obtain it? You can choose more than one.** | | | | | |
| Gloves | 106 | 14.0 | 68 | 21.5 | 0.003 |
| Hat | 124 | 16.4 | 84 | 26.5 | <0.001 |
| N95 type mask | 384 | 50.9 | 209 | 65.9 | <0.001 |
| Disposable gown | 244 | 32.3 | 146 | 46.1 | <0.001 |
| Disposable shoe protectors | 253 | 33.5 | 154 | 48.6 | <0.001 |
| Face shield | 372 | 49.3 | 194 | 61.2 | <0.001 |
| Clear protective glasses | 183 | 24.2 | 126 | 39.7 | <0.001 |
| Special protective closed suit | 448 | 59.3 | 190 | 59.9 | 0.9 |
| Biocidal hydroalcoholic solution | 80 | 16.0 | 52 | 21.5 | 0.07 |
| Disposable surgical mask | 160 | 21.2 | 92 | 29.0 | 0.006 |
| I've had adequate and sufficient PPE | 109 | 21.8 | 36 | 14.9 | 0.03 |

Data represent equipment that respondents were reporting as unavailable.

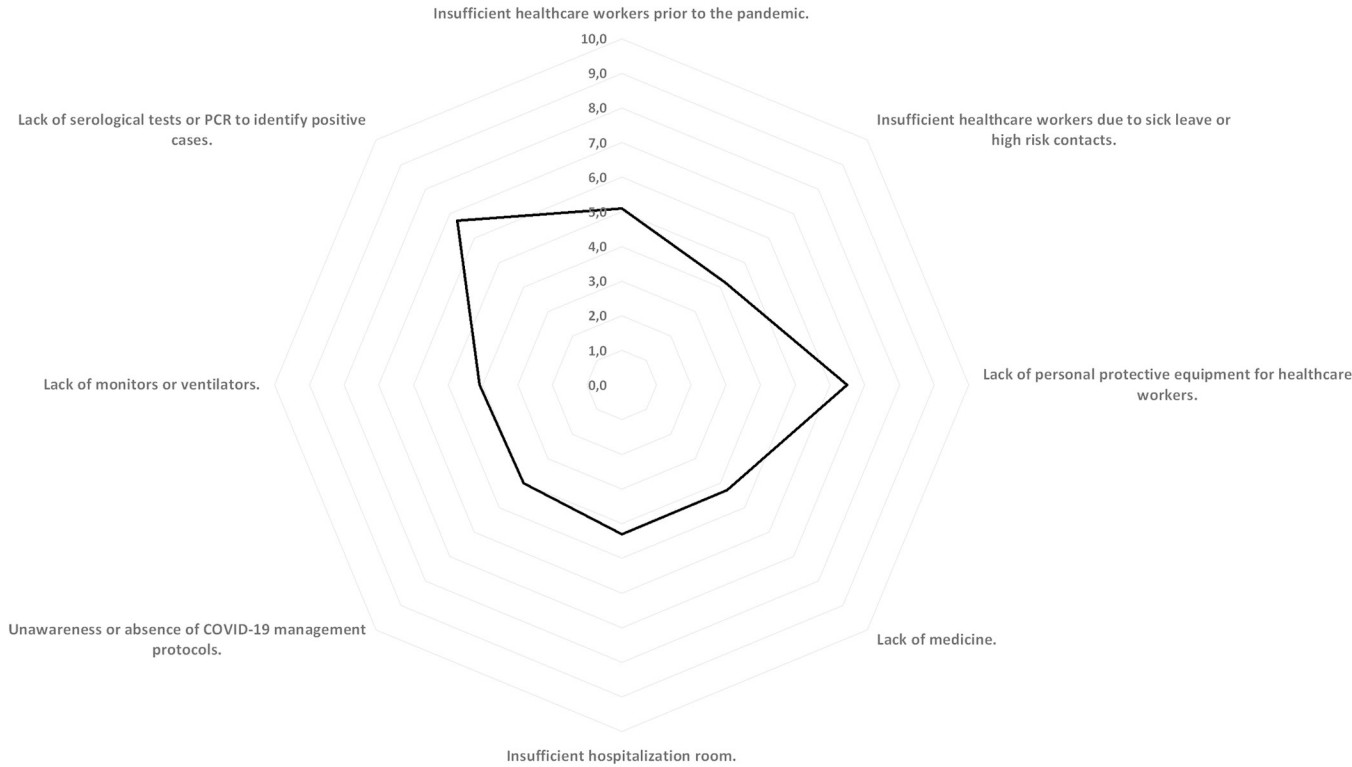

**Fig 1. Main obstacles when caring for patients with COVID-19.**

polymerase chain reaction (RT-PCR) testing for SARS-CoV-2 and isolated the affected staff members.

**Table 4. Training for correct use of PPE and use of protocols for the care of COVID-19 patients.**

| | Doctor | | Nurse | | Nursing Assistant | | Others | | p |
|---|---|---|---|---|---|---|---|---|---|
| | n = 534/1082 (49,4%) | | n = 263/1082 (24,3%) | | n = 171/1082 (15,8%) | | n = 114/1082 (10,5%) | | |
| | n | % | n | % | n | % | n | % | |
| **Have you received any training on how to use personal protective equipment?** | | | | | | | | | |
| Yes, it was enough | 126 | 32.1 | 125 | 48.3 | 74 | 43.4 | 38 | 39.2 | 0.003 |
| Yes, but it was insufficient and I would like to receive clearer information | 177 | 45.2 | 85 | 32.8 | 65 | 38 | 43 | 44.3 | |
| I don't have enough training | 89 | 22.7 | 49 | 18.9 | 32 | 18.7 | 16 | 16.5 | |
| **What type of standardized protocols or guidelines for the care of patients with suspected respiratory illness or COVID-19 are used in your center?** | | | | | | | | | |
| I don't know or we don't have protocols for common use | 50 | 9.4 | 10 | 3.8 | 20 | 11.7 | 6 | 5.3 | 0.005 |
| Private protocols exclusive to my center | 91 | 17.0 | 67 | 25.5 | 38 | 22.2 | 30 | 26.3 | |
| National Protocols | 287 | 53.7 | 145 | 55.1 | 78 | 45.6 | 60 | 52.6 | |
| Guides to global health organizations, societies or institutions abroad | 106 | 19.9 | 41 | 15.6 | 35 | 20.5 | 18 | 15.8 | |
| **If you have protocols or guidelines, what do you consider to be the main obstacle to their implementation? You can choose more than one.** | | | | | | | | | |
| Lack of habit for using protocols or unawareness of it. | 119 | 22.3 | 80 | 30.4 | 49 | 28.7 | 43 | 37.7 | 0.002 |
| It is not possible to follow them due to the limitations of the center | 281 | 52.6 | 84 | 31.9 | 59 | 34.5 | 52 | 45.6 | <0.001 |
| They change frequently and I can't keep up | 86 | 16.1 | 70 | 26.6 | 31 | 18.1 | 24 | 21.1 | 0.005 |
| I have no obstacle to applying the protocols | 67 | 27.9 | 92 | 37.6 | 64 | 38.1 | 27 | 30.7 | 0.07 |

In total, 276 (25.5%) professionals had to undergo a SARS-CoV-2 test on their own (not arranged by the workplace) and had to continue working until the result was obtained. Most of these professionals worked in public hospitals (168, 60.9%), followed by private hospitals (67, 24.3%) and primary care (17, 6.2%) (p>0.05).

Moreover, 346 (32%) participants who had close contact with a patient suspected or confirmed to have COVID-19 without adequate protective measures said that they had to continue working in the absence of symptoms. In total, 736 (68%) participants were able to maintain preventative isolation, although there were different criteria regarding the duration of isolation (7, 14, and 21 days according to different protocols).

## Qualitative analysis

Most of the participants focused on (1) the lack of PPE and the reuse of this equipment: *"Reuse of disposable gowns and smocks, they are washed and reused, the N95 mask has to last us a week;"* (2) the emotional overload caused by the pandemic in staff members, for which they lacked the necessary preparation: *"the fear, and all the mental and emotional affectation by the situation;"* (3) and in the independent performance of procedures (especially observed among general practitioners), which would have been carried out under supervision under normal conditions: *"The attending physicians, specialists, the majority of them are absent for fear of contagion and lack of PPE. Resident physicians are often making decisions based on our criteria due to lack of supervision by specialists."* Table 5 contains a summary of comments from the participants.

## Discussion

This study identified severe PPE shortages, insufficient training regarding infection prevention and PPE usage, and lack of readily available testing and isolation protocols for healthcare

**Table 5. Verbatims from the healthcare workers regarding the work conditions during the outbreak.**

| |
|---|
| *In my institution we have no personal protective equipment, we have colleagues infected due to this (18)* |
| *It takes 10–12 days to get PCR test results (11)* |
| *Testing for all health staff is missing, regardless of direct contact with patient positive for COVID-19 (10)* |
| *Transportation, we have no way to get to work (7)* |
| *We have no protocols, or we don't know about them (5)* |
| *Lack of cohesion between national and foreign guidelines (4)* |
| *Lack of coordination between the healthcare staff. (4)* |
| *An obstacle is the anxiety that is being generated in health professionals to provide a service. (4)* |
| *They do not give us personal protective equipment to the point that it is necessary to buy it.(3)* |
| *There is no education for the general population, they could be a possible source of infection (3)* |
| *No follow-up of asymptomatic cases who walk around freely (2)* |
| *Misclassification due to continuous changes of protocols. (2)* |
| *Healthcare support staff with limited training in this type of infection. (2)* |
| *Occupational physicians do not put themselves in our shoes. . . I was told that young people don't die from COVID-19 and I had to work until I had symptoms. . . (2)* |
| *We are not a high complexity emergency department and we do not handle mechanical ventilation. I had contact with a patient, and I was tested for COVID-19. (1)* |
| *I can notice that the protective equipment supplied is insufficient and we have been buying mostly masks and goggles from our own pockets. The use of alcohol is very limited, they restrict it, which I think is dangerous. I note that the nursing assistants are the most vulnerable to infection, they are not well protected. (1)* |
| *Difficult surveillance of patients in rural areas because there are very distant communities and due to the emergency situation, it is difficult to have transportation to access these areas (1)* |

The numbers in parentheses indicate the number of times the same idea was repeated.

workers in Ecuador, Brazil, and Colombia. Owing to this, many healthcare professionals have contracted COVID-19 since the initial outbreak in late 2019 [24]. It is not possible to determine the number of Latin American professionals infected in the course of caring for patients with COVID-19. Particularly, healthcare workers cited difficulties in undergoing PCR tests and the breakdown of the PPE supply chain to be their two biggest concerns. With South America on the brink of a potential second outbreak, health authorities must implement substantial changes to ensure an adequate health system response to the challenge posed by the COVID-19 pandemic.

Approximately three-quarters of the participants felt that they did not have the necessary resources to adequately care for patients with COVID-19. This was most prevalent in workplaces with specialized units (ICUs, hospital wards, or emergency departments), particularly in public hospitals. Although most centers (public and private) had protocols for caring for patients with COVID-19, the majority of the participants reported that they did not know how to implement these protocols or reported that there were significant shortcomings that prevented their implementation.

During an epidemic, the development of infection in healthcare professionals negatively impacts the capacity to treat patients, staff morale [7, 25], and public confidence. Therefore, healthcare professionals must be adequately protected. A recent study reported that appropriate PPE use could reduce the risk of contagion, even during aerosol-generating procedures [14].

In this study, only 2 out of 10 professionals who performed high-risk procedures reported that they had adequate PPE in their workplace. This study also found that the perceived needs of professionals for PPE are not always in line with the real needs for the task at hand. Nearly half of the professionals who did not perform aerosol-generating procedures reported a lack of protective eyewear or special protective suits. This result highlights a lack of dissemination of clear information; with appropriate information, professionals can appease their fears about caring for patients with COVID-19. This study highlights the lack of preparedness among healthcare personnel to protect themselves from possible infection, which could be one of the causes for the increase in the number of infected professionals.

Our results are similar to those reported by the STREPRIC group, who found that less than half of the professionals had received specific training in using PPE in a sample of family doctors in Spain. This lack of training contributes to insecurity and greater psychological distress [13]. This premise can be especially accentuated in intensive care physicians who lack the resources necessary to care for patients with COVID-19 or for self-protection to avoid infection, which can damage morale [26]. In this study, 12.6% of the participants worked in ICUs and 29.9% worked in emergency departments. This group needs to be well protected so that the capacity of treating patients can be maintained throughout the pandemic. This study reports a novel finding: there are a number of professionals who supply their own PPE and/or undergo a PCR test on their own. Up to 3 out of 5 of public health professionals had to undergo a PCR test on their own and had to continue working as long as they did not present with symptoms or obtained the test result. The reasons for this were not analyzed, but this could be due to a lack of resources and the distress of professionals of becoming ill or infecting their loved ones. Future research on this aspect is warranted.

The results obtained in this study differ in some respects from those in other continents, which could be due to differences in the health systems of the participating countries. Furthermore, the economic and social contexts and the health system model in Latin America may explain some of our findings, such as the lack of access to different PPE equipment and diagnostic methods for both professionals and patients.

Globally, a high number of health professionals have been infected with SARS-CoV-2 [27]. This may be due to several factors: the contagion rate at the beginning of the pandemic [28], the lack of protocols and training regarding the efficacious use of PPE [29], and the lack of equipment to protect against the infection risk inherent to healthcare activities [30]. The findings of this study are in agreement with those of others suggesting that the scarcity and reuse of PPE and lack of training may be the cause of the high number of healthcare workers infected worldwide [9]. However, this study shows that the fear of contagion influences professionals' perception as to what equipment was actually required.

The health emergency resulting from the COVID-19 pandemic took the health systems of Latin American countries and the world by surprise. However, the response must be reviewed to strengthen the supply chain, enhance international collaboration, and establish action plans in conjunction with international organizations to deal with possible future outbreaks and new epidemics. This review must cover training programs, both in terms of basic medical training and specialist training. Furthermore, the training delivered in the epidemiology and public health fields should be reviewed. Healthcare professionals and institutions should examine ways to strengthen the PPE supply chain.

This study has several limitations. First, although the study sample corresponded to different hospital centers in different cities of the participating countries, sampling was non-random. The sample is not completely homogeneous, as there are differences in the proportions of professional groups for each country. Second, information should be collected from professionals with COVID-19 to obtain feedback from a patient perspective; this could provide relevant information for the health system. Third, since this was a cross-sectional study, it is not possible to follow the evolution of and the limitations and requirements faced by frontline professionals. Lastly, no pilot test of the survey was conducted due to the urgent nature of the COVID-19 pandemic.

Healthcare professionals in Latin America may face more difficulties than those from other continents, mainly Europe, Asia, and North America. In particular, access to PCR tests in case of close contact with a person with COVID-19 without appropriate PPE and inadequacies related to diagnosing and treating patients with COVID-19 appear to be significant issues. The availability of PPE is essential for the health system to continue functioning and cope with the pandemic; yet, the general perception of healthcare workers is that they have not been able to access adequate PPE to protect themselves from COVID-19. Healthcare workers must feel protected and be aware of the proper PPE needed in each scenario. If this is not the case, diminished work morale could harm the professional's resilience to endure the pandemic. Policymakers should ensure access to diagnostic testing and adequate PPE for healthcare workers. Technical and logistical difficulties should be addressed in the event of a future outbreak by learning from our experience with the COVID-19 pandemic. Further studies are needed in subsequent phases of the pandemic to assess and compare the learnings, capacity, and adaptability of the health systems in South America and address any further concerns.

## Conclusions

In conclusion, this study sought to identify the main difficulties and obstacles faced by frontline professionals caring for patients with COVID-19 in three Latin American countries. This pandemic has presented unprecedented challenges, and difficulties have been encountered worldwide. Developing countries with economic difficulties face additional challenges in this regard. However, for healthcare professionals to provide adequate care to patients with and without COVID-19 during the pandemic, professionals should feel physically and mentally prepared. It is important for authorities to provide an efficient supply chain, up-to-date

protocols, and clear information. Our study has highlighted some areas that need to be improved for dealing with further waves of the COVID-19 outbreak and potential future pandemics and epidemics.

## Supporting information

**S1 File. Survey conducted in the study.** Anonymous survey administered to the participants. (DOCX)

**S1 Table. Resources for appropriate diagnosis and treatment of COVID-19 patients in several settings.**
(DOCX)

**S2 Table. Unavailability of personal protective equipment as reported by each professional group.**
(DOCX)

## Acknowledgments

The authors would like to thank all the healthcare workers who voluntarily responded to this study and all the rest of the team in the frontline.

## Author Contributions

**Conceptualization:** Jimmy Martin-Delgado, Eduardo Viteri, Piedad Serpa.

**Data curation:** Jimmy Martin-Delgado, Aurora Mula, Piedad Serpa, Gloria Pacheco, Diana Prada, Daniela Campos de Andrade Lourenção, Patricia Campos Pavan Baptista, Gustavo Ramirez, Jose Joaquin Mira.

**Formal analysis:** Jimmy Martin-Delgado, Aurora Mula, Jose Joaquin Mira.

**Investigation:** Jimmy Martin-Delgado.

**Methodology:** Jimmy Martin-Delgado, Aurora Mula, Jose Joaquin Mira.

**Project administration:** Jose Joaquin Mira.

**Supervision:** Jose Joaquin Mira.

**Writing – original draft:** Jimmy Martin-Delgado, Eduardo Viteri, Piedad Serpa, Jose Joaquin Mira.

**Writing – review & editing:** Jimmy Martin-Delgado, Eduardo Viteri, Piedad Serpa, Gloria Pacheco, Diana Prada, Daniela Campos de Andrade Lourenção, Patricia Campos Pavan Baptista, Gustavo Ramirez, Jose Joaquin Mira.

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
