## [Decision Letter · Decision Letter 0]

10 Sep 2020

PONE-D-20-25186

Personal protective equipment, diagnostic and treatments facilities for COVID-19 patients. A cross-sectional study in Brazil, Colombia and Ecuador.

PLOS ONE

Dear Dr. Martin-Delgado,

Thank you for submitting your manuscript to PLOS ONE. After careful consideration, we feel that it has merit but does not fully meet PLOS ONE’s publication criteria as it currently stands. Therefore, we invite you to submit a revised version of the manuscript that addresses the points raised during the review process.<please by="" manuscript="" revised="" submit="" your="">

Please include the following items when submitting your revised manuscript:</please>

We look forward to receiving your revised manuscript.

Kind regards,

Khin Thet Wai, MBBS, MPH, MA (Population & Family Planning Resear

Academic Editor

PLOS ONE

Journal Requirements:

2. Please provide further details on sample size and power calculations. Please provide approximate numbers of health workers (especially doctors and nurses) and treatment facilities for COVID-19 in the 3 countries. Please discuss whether your sample size would be sufficiently representative of your target population.

4.

We note that you have indicated that data from this study are available upon request. PLOS only allows data to be available upon request if there are legal or ethical restrictions on sharing data publicly. For information on unacceptable data access restrictions, please see http://journals.plos.org/plosone/s/data-availability#loc-unacceptable-data-access-restrictions.

5. Please amend your authorship list in your manuscript file to include author Stefany Pacheco

6. Please amend the manuscript submission data (via Edit Submission) to include author Gloria Pacheco

Additional Editor Comments (if provided):

English language correction by the well-recognized editing service is deemed necessary.

Methodological weaknesses as pointed out by the reviewer should be taken into account to strengthen scientific rigour.

Reviewers' comments:

Reviewer's Responses to Questions

**Comments to the Author**

1. Is the manuscript technically sound, and do the data support the conclusions?

Reviewer #1: Yes

Reviewer #2: Yes

2. Has the statistical analysis been performed appropriately and rigorously? 

Reviewer #1: No

Reviewer #2: Yes

3. Have the authors made all data underlying the findings in their manuscript fully available?

Reviewer #1: Yes

Reviewer #2: Yes

4. Is the manuscript presented in an intelligible fashion and written in standard English?

Reviewer #1: No

Reviewer #2: Yes

5. Review Comments to the Author

Reviewer #1: This is a survey of healthcare workers (HCW) in South America aimed at evaluating PPE shortages and related issues arising from the COVID-19 pandemic. It is true that such data from South America is scarce, and as such, an important gap is being addressed by the authors. I commend them on this tremendous effort in the midst of a raging pandemic and wish them well.

I have listed some major and minor points for consideration by the authors. Overall, there is some unique data presented here, and I believe it would be of significant interest to the healthcare community in Latin America and globally.

The methodology is appropriate for a pandemic, but does have some limitations that should be clearly articulated. Overall, the methods section needs to be clearer and reported according to one of the many published checklists or guidelines (I have provided citations below, but the authors may choose to use something else).

I do not believe it is appropriate or desirable to report inferential statistics with this survey. Descriptive statistics with a focus on the shortcomings in the pandemic response (PPE availability, lack of education, lack of COVID-19 testing for HCWs) would be my recommendation.

Specific points listed below.

Major

1.I fully understand that English may not be the first language for all or most of the authors. Respectfully, the grammar and general usage of the English language throughout this manuscript must be improved prior to consideration for publication. I have pointed out a small sample of these in the Minor comments to assist you in identifying areas for improvement.

2.Methods:

- Please provide a more detailed methodology section. There are lots of items missing. I recommend the use of a tool such as described in this paper by the ACCADEMY group: A guide for the design and conduct of self-administered surveys of clinicians. Karen E.A. Burns, Mark Duffett, Michelle E. Kho, Maureen O. Meade, Neill K.J. Adhikari, Tasnim Sinuff, Deborah J. Cook. CMAJ Jul 2008, 179 (3) 245-252; DOI: 10.1503/cmaj.080372

- there is a nice checklist in this paper that may also be useful: Artino AR Jr, Durning SJ, Sklar DP. Guidelines for Reporting Survey-Based Research Submitted to Academic Medicine. Acad Med. 2018;93(3):337-340. doi:10.1097/ACM.0000000000002094

- it is acceptable to clearly state this is a survey conducted by non-random, snowball sampling and internal pilot testing without a fully validated survey tool (appropriate given the urgency of the pandemic)

3.Results:

- The columns in each of the tables are different. Please pick one categorisation and stick to it. Where necessary, the other categorisations may be used in the text and listed in the appendix.

- My preference would be for predominant reporting of descriptive statistics without an emphasis on inference. This survey is simply not designed to answer any inferential questions, though it is very tempting to calculate p values when you have a reasonably large dataset.

4.Free text comments

- please consider listing more of the free text comments in a separate Table- these are very powerful and in my opinion, some of the most important findings from your survey. Please use this platform to help colleagues without a voice be heard.

5.Discussion:

- the first paragraph of the discussion should list all the key findings from the study, rather than re-state the aim which has already been done. I would say that the severe PPE shortages, lack of readily available testing and preventative isolation for HCWs and inadequacy of PPE and infection prevention training for HCWs.

- I would also add a “Conclusions” paragraph at the end of the Discussion which again stresses the key messages from the survey and one or two key learning points or future directions.

- please emphasise the need for governments and policymakers to be aware of this sort of data and for professionals bodies to agitate for change.

Minor

1.When reporting large numbers, please use commas as separators for every 3 places (e.g. 1,000,000).

2.P9 L 59- remove today and just state the date on which the numbers were updated.

3.P9 L 68- “outpointed” is used incorrectly. Please change to “demonstrated”

4.P10 L95- Please state a primary objective and one or more secondary objectives

5.P11 L107- “telematic” is not a recognised word- I would suggest “teleconferencing”.

6.P12 L143-145- I understand that due to the language the WHO has been using, you have chosen to separate the groups based on aerosol generating procedures- however, there is increase evidence that simple things like breathing and talking are aerosol generating, and that the aerosol/droplet dichotomy is an oversimplification. Some acknowledgement of this would be appropriate, even if you don’t change your actual methods.

7.P13 L147-153- What surveying software was used? Google forms, Survey Monkey etc

Was the number of participants you wanted to recruit prospectively determined?

Your sampling technique is non-random (which you have mentioned in the abstract but not in the Methods section). In this setting, statistical significance has limited meaning. I would suggest you state that you wanted a large enough sample to be able to make some generalisations, rather than for the sole purpose of generating small enough p values.

8.Table 5- currently not very clear as to what the table is reporting. I’m assuming that the percentages are for equipment that respondents are reporting as unavailable.

Reviewer #2: The author(s) choose appropriate web based data collection method and inclusion of qualitative portion can add more information. The study consist appropriate categories of health professional and consist both public and private setting. But weak in statistical analysis.

6. PLOS authors have the option to publish the peer review history of their article (what does this mean?). If published, this will include your full peer review and any attached files.

Reviewer #1: **Yes: **Dr Mahesh Ramanan

Reviewer #2: **Yes: **Dr Hla Hla Win

---

## [Author Response · Author response to Decision Letter 0]

8 Oct 2020

Dear Editor, 

We appreciate very much the work done by the reviewers, particularly in this period we assume that is an additional effort as we are well aware of the difficulties arising from the COVID outbreak. Many thanks. Also, we appreciate the suggestions. We are sure they improve this work. We have followed your directions and included a manuscript with tracked changes, and a revised manuscript. In this letter, we comment one by one on these suggestions.

Yours faithfully 

The authors

Editorial comments: 

We have revised the manuscript in order to meet PLOS One style requirements.

2. Please provide further details on sample size and power calculations. Please provide approximate numbers of health workers (especially doctors and nurses) and treatment facilities for COVID-19 in the 3 countries. Please discuss whether your sample size would be sufficiently representative of your target population.

Further details and information from national and international official statistics sources have been added, and a comment made in the methods section. 

Attached you´ll find Editage certificate. Thank you for the advice provided. 

4. We note that you have indicated that data from this study are available upon request. 

We have uploaded our data set in an open access file repository:

Mira JJ. PPE, diagnostic and treatment facilities for COVID-19 in South America 2020. doi:10.17605/OSF.IO/4EG8R.

Also, the study is publicly available at ClinicalTrials: NCT04486404

5. Please amend your authorship list in your manuscript file to include author Stefany Pacheco

We have amended our authorship list via the Edit Submission and included author Gloria Pacheco. 

Reviewer comments:

1. Please provide a more detailed methodology section. There are lots of items missing. 

We have followed the ACCADEMY group guideline for design and conduct of self-administered surveys of clinicians, and we appreciate the advice given by the reviewer. We have made the necessary corrections and provided a more detailed methods section.

2. The columns in each of the tables are different. Please pick one categorisation and stick to it. Where necessary, the other categorisations may be used in the text and listed in the appendix.

We have taken under consideration reviewer advice and included two of the original tables as a supplementary file to be available online. More precisely, former Table 2 and 4. We acknowledge, that having different columns to display information may be confusing but we consider this to be necessary as relevant information is provided. We consider that tables comparing public and private resources available to healthcare workers, differences of Personal Protective Equipment at disposal of healthcare workers who performed high risk of contagion procedures and finally, training for correct use of PPE and use of protocols for healthcare workers to be important, as they align with the recommendations made to managers and policy makers. 

3. My preference would be for predominant reporting of descriptive statistics without an emphasis on inference. This survey is simply not designed to answer any inferential questions, though it is very tempting to calculate p values when you have a reasonably large dataset.

We agree with the reviewer and we have made changes to our methods section. Even though we have decided to keep Chi Square analysis in some categories, this could be removed if editorial and reviewers consider it necessary. 

4. Please consider listing more of the free text comments in a separate Table- these are very powerful and in my opinion, some of the most important findings from your survey. Please use this platform to help colleagues without a voice be heard.

We cannot agree more with you, relevant information was provided by means of the free text comments, we received a total of 365. We have included Table 5, which lists more extracts and the number of times a similar idea was expressed by respondents. 

5. the first paragraph of the discussion should list all the key findings from the study, rather than re-state the aim which has already been done. I would say that the severe PPE shortages, lack of readily available testing and preventative isolation for HCWs and inadequacy of PPE and infection prevention training for HCWs. 

We have made the necessary corrections and included key findings in the first paragraph of the discussion. 

6. I would also add a “Conclusions” paragraph at the end of the Discussion which again stresses the key messages from the survey and one or two key learning points or future directions.

A “conclusion” paragraph was added at the end of the discussion and also key points for managers and policy makers were included. Hopefully this research can provide feedback and future directions for governments in South America. 

7. I understand that due to the language the WHO has been using, you have chosen to separate the groups based on aerosol generating procedures- however, there is increase evidence that simple things like breathing and talking are aerosol generating, and that the aerosol/droplet dichotomy is an oversimplification. Some acknowledgement of this would be appropriate, even if you don’t change your actual methods.

We recognize this dichotomy could be an oversimplification, but we can also agree on that some procedures poses an inherent risk that could only be performed in safe spaces of a hospital. This is why we have included the following phrase in our methods section “This distribution was done considering the increased risk of infection associated with some COVID-19 patient care practices as opposed to others, which normally advise differentiated personal protective equipment needs.” Which is also why, we have considered as important to preserve the table comparing both circumstances, where protocols differ as which PPE are necessary for each of them. We believe, it is also important to communicate how this increased perception of higher risk and exposure of HCW (reported by other studies) can modify their demands. One example of this, is how 60% of participants who do not perform high risk procedures reported as unavailable special protected closed suits, when it is not necessary, according to current protocols. 

Minor comments

We have made the necessary corrections and copyedited our manuscript using Editage services. 

We have included two objectives in the last paragraph of the introduction. 

We have included details about Surveymonkey as the platform used to conduct the study. 

More details about the sample and its size has been added to the methods.

---

## [Decision Letter · Decision Letter 1]

14 Oct 2020

PONE-D-20-25186R1

Personal protective equipment, diagnostic and treatments facilities for COVID-19 patients. A cross-sectional study in Brazil, Colombia and Ecuador.

PLOS ONE

Dear Dr. Martin-Delgado,

Thank you for submitting your manuscript to PLOS ONE. After careful consideration, we feel that it has merit but does not fully meet PLOS ONE’s publication criteria as it currently stands. Therefore, we invite you to submit a revised version of the manuscript that addresses the points raised during the review process.<please by="" manuscript="" revised="" submit="" your="">

Please include the following items when submitting your revised manuscript:</please>

We look forward to receiving your revised manuscript.

Kind regards,

Khin Thet Wai, MBBS, MPH, MA (Population & Family Planning Resear

Academic Editor

PLOS ONE

Reviewers' comments:

Reviewer's Responses to Questions

**Comments to the Author**

1. If the authors have adequately addressed your comments raised in a previous round of review and you feel that this manuscript is now acceptable for publication, you may indicate that here to bypass the “Comments to the Author” section, enter your conflict of interest statement in the “Confidential to Editor” section, and submit your "Accept" recommendation.

Reviewer #1: All comments have been addressed

2. Is the manuscript technically sound, and do the data support the conclusions?

Reviewer #1: Yes

3. Has the statistical analysis been performed appropriately and rigorously? 

Reviewer #1: Yes

4. Have the authors made all data underlying the findings in their manuscript fully available?

Reviewer #1: Yes

5. Is the manuscript presented in an intelligible fashion and written in standard English?

Reviewer #1: Yes

6. Review Comments to the Author

Reviewer #1: Thank you for addressing my comments and congratulations on an important piece of work.

I have a small number of minor revisions that I recommend:

L40: change “it was not randomized” to “sampling was non-random”

L56-59: update COVID case numbers and deaths from latest WHO Sitrep

L153: change “It was prepared using…” to “We used..” for simplicity

Table 3, L212: “Did not performed” should read “Did not perform”

Discussion

L272-273: The first two sentences do not belong here. They should either be deleted or moved later into the discussion. I think you should start with your major findings i.e. “This study has identified severe PPE shortages, insufficient training in infection prevention and PPE usage, and lack of readily available testing and isolation protocols for healthcare workers in Ecuador, Brazil and Colombia…..”

L346: “sampling was non-random”

L375: drop this sentence “Otherwise, responsiveness is questioned”.

L376: “Our study has highlighted some areas that need to be improved **for further waves of COVID-19 and potential future pandemics and epidemics**.”

7. PLOS authors have the option to publish the peer review history of their article (what does this mean?). If published, this will include your full peer review and any attached files.

Reviewer #1: **Yes: **Mahesh Ramanan

---

## [Author Response · Author response to Decision Letter 1]

15 Oct 2020

Dear Editor, 

We appreciate very much the work done by the reviewer, particularly in this period we assume that is an additional effort as we are well aware of the difficulties arising from the COVID outbreak. Many thanks. Also, we appreciate the suggestions. We are sure they improve this work. We have followed your directions and included a manuscript with tracked changes, and a revised manuscript. In this letter, we comment one by one on these suggestions.

Yours faithfully 

The authors

Reviewer comments:

Minor comments

L40: change “it was not randomized” to “sampling was non-random”

We have unified the term to “non-random” across the manuscript. 

L56-59: update COVID case numbers and deaths from latest WHO Sitrep

We have updated COVID cases and deaths according to WHO Dashboard, and changed the citation. 

L153: change “It was prepared using…” to “We used..” for simplicity

We have included your suggestion. We agree, on simplicity. 

Table 3, L212: “Did not performed” should read “Did not perform”

We have corrected the table. 

Discussion

L272-273: The first two sentences do not belong here. They should either be deleted or moved later into the discussion. I think you should start with your major findings i.e. “This study has identified severe PPE shortages, insufficient training in infection prevention and PPE usage, and lack of readily available testing and isolation protocols for healthcare workers in Ecuador, Brazil and Colombia…..”

We have modified the first paragraph of the Discussion. Thank you for your advice. 

L375: drop this sentence “Otherwise, responsiveness is questioned”.

The sentence has been removed.

L376: “Our study has highlighted some areas that need to be improved **for further waves of COVID-19 and potential future pandemics and epidemics**.”

We have included reviewer suggestion.

---

## [Editor Report · Decision Letter 2]

19 Oct 2020

PONE-D-20-25186R2

Personal protective equipment, diagnostic and treatments facilities for COVID-19 patients. A cross-sectional study in Brazil, Colombia and Ecuador.

PLOS ONE

Dear Dr. Martin-Delgado,

Thank you for submitting your manuscript to PLOS ONE. After careful consideration, we feel that it has merit but does not fully meet PLOS ONE’s publication criteria as it currently stands. Therefore, we invite you to submit a revised version of the manuscript that addresses the points raised during the review process.<please by="" manuscript="" revised="" submit="" your=""></please>

<please by="" manuscript="" revised="" submit="" your="">Please include the following items when submitting your revised manuscript:</please>

We look forward to receiving your revised manuscript.

Kind regards,

Khin Thet Wai, MBBS, MPH, MA (Population & Family Planning Resear

Academic Editor

PLOS ONE

Additional Editor Comments (if provided):

To correct minor grammatical errors throughout the manuscript.

---

## [Author Response · Author response to Decision Letter 2]

26 Oct 2020

Dear Editor, 

We appreciate very much the work done by the reviewers and editorial office, particularly in this period we assume that is an additional effort as we are well aware of the difficulties arising from the COVID outbreak. Many thanks. Also, we appreciate the suggestions. We are sure they improve this work. We have followed your directions and included a manuscript with tracked changes, and a revised manuscript. In this letter, we comment one by one on these suggestions.

Yours faithfully 

The authors

Reviewer comments:

Minor comments

To correct minor grammatical errors throughout the manuscript.

We have provided a new version that has been copyedited by Editage and supplied the certificate of editing.

While revising your submission, please upload your figure files to the Preflight Analysis and Conversion Engine (PACE) digital diagnostic tool.

We have followed PLOS ONE requirements and used PACE to ensure that.

---

## [Editor Report · Decision Letter 3]

29 Oct 2020

Availability of personal protective equipment and diagnostic and treatment facilities for healthcare workers involved in COVID-19 care: A cross-sectional study in Brazil, Colombia, and Ecuador.

PONE-D-20-25186R3

Dear Dr. Martin-Delgado,

We’re pleased to inform you that your manuscript has been judged scientifically suitable for publication and will be formally accepted for publication once it meets all outstanding technical requirements.

Kind regards,

Khin Thet Wai, MBBS, MPH, MA (Population & Family Planning Resear

Academic Editor

PLOS ONE

Additional Editor Comments (optional):

All requirements inclusive of language correction have been fully addressed.
---

## [Editor Report · Acceptance letter]

3 Nov 2020

PONE-D-20-25186R3 

Availability of personal protective equipment and diagnostic and treatment facilities for healthcare workers involved in COVID-19 care: A cross-sectional study in Brazil, Colombia, and Ecuador 

Dear Dr. Martin-Delgado:

I'm pleased to inform you that your manuscript has been deemed suitable for publication in PLOS ONE. Congratulations! Your manuscript is now with our production department. 

Kind regards, 

on behalf of

Dr. Khin Thet Wai 

Academic Editor

PLOS ONE